# Translational initiation in *E. coli* occurs at the correct sites genome-wide in the absence of mRNA-rRNA base-pairing

**Kazuki Saito[1], Rachel Green[1,2], Allen R Buskirk[1]\***

[1]Department of Molecular Biology and Genetics, Johns Hopkins University School of Medicine, Baltimore, United States; [2]Howard Hughes Medical Institute, Johns Hopkins University School of Medicine, Baltimore, United States

**Abstract** Shine-Dalgarno (SD) motifs are thought to play an important role in translational initiation in bacteria. Paradoxically, ribosome profiling studies in *E. coli* show no correlation between the strength of an mRNA's SD motif and how efficiently it is translated. Performing profiling on ribosomes with altered anti-Shine-Dalgarno sequences, we reveal a genome-wide correlation between SD strength and ribosome occupancy that was previously masked by other contributing factors. Using the antibiotic retapamulin to trap initiation complexes at start codons, we find that the mutant ribosomes select start sites correctly, arguing that start sites are hard-wired for initiation through the action of other mRNA features. We show that A-rich sequences upstream of start codons promote initiation. Taken together, our genome-wide study reveals that SD motifs are not necessary for ribosomes to determine where initiation occurs, though they do affect how efficiently initiation occurs.

## Introduction

Translational initiation is a critical step in the regulation of gene expression that impacts which proteins are synthesized and to what extent. Unlike eukaryotic ribosomes, which scan from the 5'-end of messages and generally initiate at the first start codon, bacterial ribosomes can initiate at any position along an mRNA; this is a critical requirement because many bacterial mRNAs are polycistronic. Bacterial ribosomes must select the correct start codons amidst a vast excess of potential sites (AUG, GUG, and to some extent UUG) that have to be ignored. Not only does initiation determine where translation occurs (and therefore which proteins are made), in most cases the rate of initiation determines the level of protein output. In bacteria, a common strategy for regulating translation is to block ribosome recruitment to an mRNA through the action of small RNAs (*Altuvia et al., 1998*; *Majdalani et al., 1998*; *Storz et al., 2004*), small-molecule binding riboswitches (*Winkler et al., 2002*; *Mandal and Breaker, 2004*), and regulatory proteins (*Moine et al., 1990*; *Babitzke et al., 2009*).

Initiation rates vary in response to several mRNA features that determine how effectively an mRNA recruits 30S subunits to the start codon. Thermodynamically stable secondary structures surrounding the initiation site prevent 30S recruitment (*Hall et al., 1982*; *de Smit and van Duin, 1990*). The kinetics of RNA folding and unfolding are also critical (*de Smit and van Duin, 2003*; *Espah Borujeni and Salis, 2016*): some structures exist in an unfolded state for such a short period of time that 30S subunits cannot find the start codon quickly enough by diffusion alone. In several well-characterized examples, regions of single-stranded RNA known as standby-sites are found nearby, positioning 30S subunits in close proximity so that they can efficiently capture the start codon upon unfolding of the mRNA secondary structure (*de Smit and van Duin, 2003*; *Espah Borujeni et al., 2014*). Interactions of 30S subunits and single-stranded mRNA regions

**\*For correspondence:** buskirk@jhmi.edu

(especially those that are AU-rich) can be mediated through ribosome protein S1 (*Boni et al., 1991*; *Komarova et al., 2005*). Bound on the back of the 30S subunit, the S1 protein contains multiple RNA-binding domains that can recruit mRNA and melt secondary structures (*Qu et al., 2012*), facilitating hybridization of 16S rRNA with complementary mRNA sequences colloquially known as Shine-Dalgarno motifs.

Shine-Dalgarno motifs have the consensus sequence GGAGG and can base pair with as many as nine nt in the 3′ terminal sequence of 16S rRNA (ACCUCCUUA in *E. coli*) referred to as the anti-Shine Dalgarno or ASD (*Shine and Dalgarno, 1974*). Pairing of the SD-ASD sequences can recruit 30S subunits to the start codon 5–10 nt downstream (*Steitz and Jakes, 1975*). SD motifs that differ significantly from the consensus or that are positioned too close or too far from the start codon yield lower levels of initiation. Indeed, many experiments using reporter genes showed that raising the SD-ASD affinity increases protein output, demonstrating its importance for determining translation levels (*Hui and de Boer, 1987*; *Jacob et al., 1987*; *de Smit and van Duin, 1990*; *Salis et al., 2009*). In addition, the SD model serves as the foundation of practical bioengineering efforts ranging from optimizing expression of recombinant proteins to expansion of the genetic code (*Rackham and Chin, 2005*; *Salis et al., 2009*).

On the other hand, even though the ASD in 16S rRNA is almost universally conserved throughout the bacterial kingdom (*Nakagawa et al., 2010*), the percentage of genes with SD motifs varies widely between species. While well-characterized model species such as *E. coli* and *B. subtilis* have a high percentage of genes with SD motifs (54% and 78% respectively), there is little to no enrichment of SD motifs upstream of start codons in Bacteriodetes and Cyanobacteria (*Nakagawa et al., 2010*). In addition, although the majority of species in the phyla Firmicutes, Actinobacteria, and Proteobacteria have high percentages of SD-containing genes, several species have low percentages, arguing that the loss of this mechanism has occurred multiple times during evolution (*Nakagawa et al., 2010*; *Hockenberry et al., 2017*). These variations across the bacterial kingdom, despite the high conservation of the ASD element on the ribosome, raise questions as to how important the SD mechanism is for ribosome recruitment.

Ribosome profiling is a method for deep sequencing of ribosome-protected mRNA fragments that allows us to define the position and number of ribosomes bound across the transcriptome at nucleotide resolution (*Ingolia et al., 2009*). This information allows us to calculate the ribosome density on each mRNA as a proxy for the efficiency of translation initiation. In pioneering ribosome profiling studies in bacteria, the paradoxical observation was made that there is little or no correlation between the ribosome occupancy of a gene and the strength of its SD motif (calculated using thermodynamic algorithms for RNA pairing), as had been anticipated based on the SD model (*Li et al., 2014*; *Schrader et al., 2014*; *Li, 2015*; *Del Campo et al., 2015*). This surprising observation suggested that other mRNA features could effectively mask the effects of the SD correlation at the genome-wide level.

To isolate the effects of SD motifs on the global translational landscape, we expressed 16S rRNA mutants with altered (non-functional) ASD sequences, purified mutant ribosomes, and used ribosome profiling to ask how efficiently they translate each mRNA in the cell. Unlike previous studies that vary the SD motif and other mRNA-specific features, this approach allows us to specifically eliminate the SD-ASD interaction while keeping mRNA sequences and structures intact, so that we can specifically ask questions about the role the SD-ASD interaction plays in determining mRNA translation rates. Through this analysis, we observe for the first time the effects of SD motifs at the global level, revealing a linear correlation between SD strength and ribosome occupancy. We then combined our new profiling approach with retapamulin treatment to trap ribosomes at start codons (*Meydan et al., 2019*; *Weaver et al., 2019*) in order to study the role of SD motifs in selecting start codons. To our surprise, the ASD-mutant ribosomes selectively recognize the correct initiation sites as well as wild-type ribosomes, arguing that these sites are hard-wired for initiation independent of their SD-ASD pairing strength. We show that A-rich sequences recently identified by Fredrick and co-workers (*Baez et al., 2019*) are enriched at annotated start sites compared to other AUG codons in the transcriptome where initiation does not take place; these A-rich sequences are also found upstream of start codons in a wide variety of species across the bacterial kingdom. In addition, mRNA structure at annotated start sites is lower than at other AUG codons, facilitating 30S binding. Together, these studies refine our understanding of the role of SD motifs and other mRNA features in defining the proteomes of bacteria.

## Results

### Selective profiling of ribosomes with mutant ASD sequences

Studies of the role of SD motifs in promoting translation in their native contexts have been complicated by the fact that changing the sequence of an mRNA also affects other determinants of translational regulation such as its overall structure. To perturb the function of SD motifs at the global level, we developed a new approach in which we mutate the ASD in 16S rRNA, purify the mutant ribosomes, and use ribosome profiling to ask how efficiently they translate each mRNA in the cell. This strategy provides us with a genome-wide view of the function of SD motifs in interactions with the unaltered transcriptome—all of the features of an mRNA that affects its translation are maintained, thereby isolating the effects of the SD motif mutation. In this manner, we eliminate the SD-ASD interaction as a contribution to mRNA translation rates and see how translation changes across the transcriptome.

We created three 16S rRNA alleles in which the ASD is mutated (*Figure 1A*). Two of these mutants were described previously in the literature. The ASD in *specialized* (S) ribosomes was inverted from CCUCC to GGAGG in a pioneering study by de Boer who showed that although these S-ribosomes were relatively inactive on endogenous transcripts, they efficiently translate a reporter gene with a complementary SD motif (*Hui and de Boer, 1987*). In later studies, Cunningham and Chin used genetic selections to characterize additional SD-ASD pairs and improve their selectivity, creating *orthogonal* (O) ribosomes where the ASD is mutated to UGGGA (*Lee et al., 1996*; *Rackham and Chin, 2005*). Ribosomes with mutant ASD motifs (like S and O) have been used in numerous studies of protein synthesis where they selectively translate reporter genes with complementary SD motifs (*Rex et al., 1994*; *Neumann et al., 2010*; *Orelle et al., 2015*). In addition to these two ASD mutants, we constructed a third (A) with the ASD sequence AAAAA that we anticipated would bind mRNA more weakly than the O- or S-ribosomes (given that their ASD sequences are G-rich). The MS2 aptamer was inserted into these three ASD mutants to facilitate their purification as described below (*Youngman et al., 2004*; *Youngman and Green, 2005*); as a control, we also created an MS2-tagged 16S rRNA with the canonical ASD sequence (C).

These four 16S rRNA mutants were expressed from plasmids in *E. coli* MG1655 containing the normal complement of seven wild-type rRNA operons to sustain growth. Because overexpression of ASD mutants is toxic (*Jacob et al., 1987*), we induced expression for only 20–25 min during which growth rates were not affected (*Figure 1—figure supplement 1A*). Polysome profiles from the four mutants were similar (*Figure 1—figure supplement 1B*) suggesting that translation remains robust during the transient expression of MS2-tagged 16S mutants whether the ASD is intact (C) or mutated (S, O, and A). A previous study of orthogonal ribosomes suggested that altering the ASD in 16S rRNA reduces rRNA processing efficiency, leading to the accumulation of processing intermediates, but that mature rRNAs containing ASD mutations have the correct 3'-end (*Aleksashin et al., 2019*). To look for processing defects in our system, we performed RNA-seq on affinity-captured MS2-tagged rRNA without nuclease digestion. As shown in *Figure 1—figure supplement 1C*, we do not observe the accumulation of precursors with 3'-extensions or other defects in the processing of the 3'-end of 16S rRNA. This result indicates that correctly processed rRNA is produced and should be able to form mature 30S subunits.

RT-PCR with primers that distinguish endogenous 16S rRNA from the MS2-tagged mutants was used to ask whether the ASD mutants are found in actively translating polysomes (*Figure 1—figure supplement 1D*). We observed that the signal from C-ribosomes is equally strong in the lysate, light, and heavy polysome fractions. In contrast, the signal from the three ASD mutants is present but weaker in the polysome fractions than in the lysate. These data show that although ribosomes with mutant ASDs can engage in translation, their activity is impaired, consistent with earlier studies. Keeping this in mind, we focus our analyses not on their absolute activity but on their selectivity, asking which mRNAs they translate better than other mRNAs.

To purify mutant ribosomes, we employed a method previously developed for in vitro biochemical studies of ribosomes with lethal mutations (*Youngman et al., 2004*; *Youngman and Green, 2005*): the MS2 aptamer was fused to helix 6 of 16S rRNA allowing us to capture mutant ribosomes through their interaction with the MS2 coat protein (*Figure 1B*). To avoid pulling down wild-type ribosomes bound to the same mRNA as mutant ribosomes, we first treated cell lysates with RNase T1 to collapse polysomes to monosomes prior to isolating MS2-tagged ribosomes. RT-PCR reveals

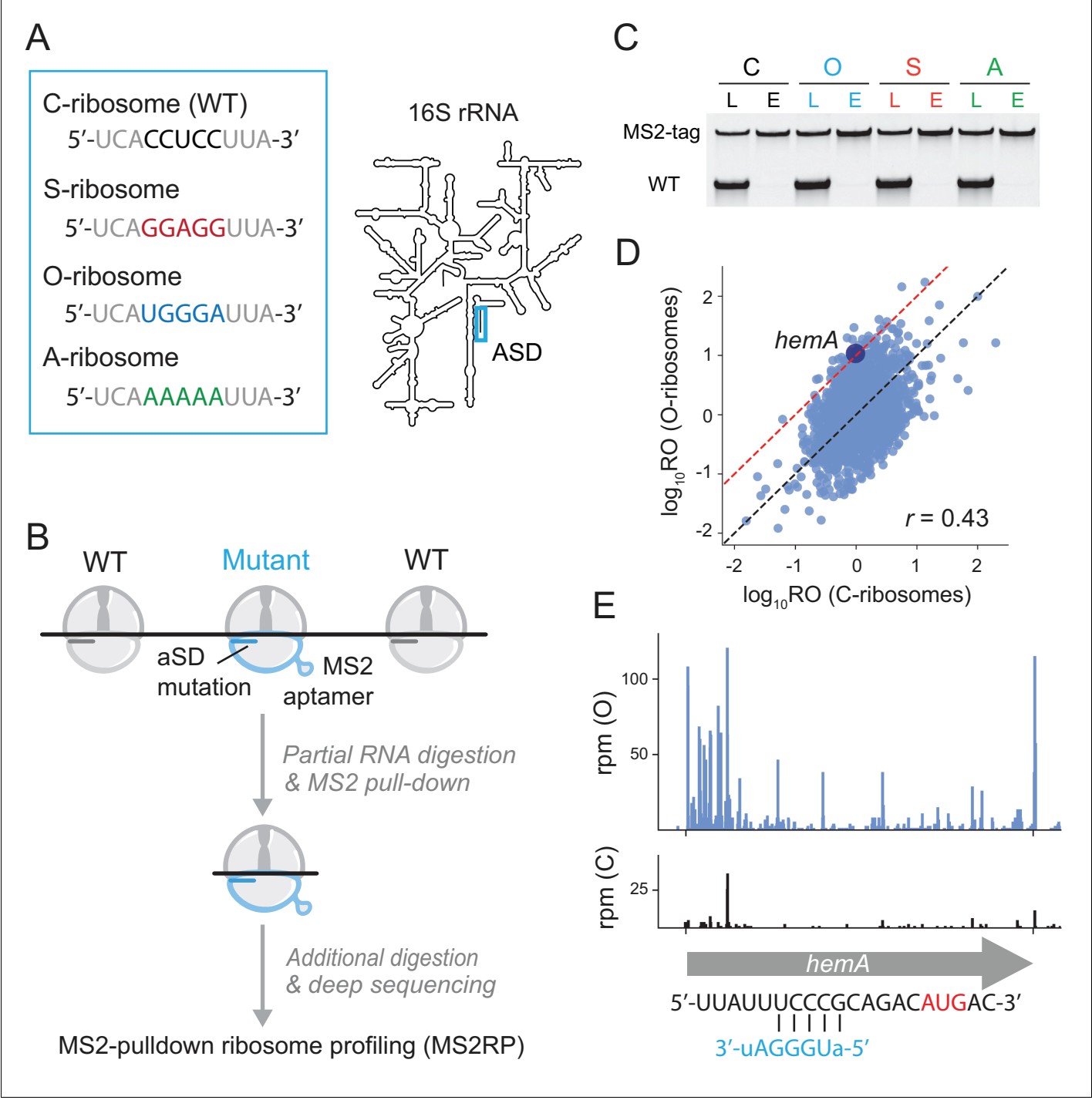

**Figure 1.** Capturing the role of SD motifs by MS2RP. (**A**) ASD mutations at the 3'-end of 16S rRNA are highlighted in color. (**B**) Schematic of MS2RP: polysomes are collapsed to monosomes by RNase T1 digestion, MS2-tagged monosomes are pulled down with the MS2 coat-protein, and mRNA is fully digested to yield ribosome footprints that are subjected to deep sequencing. (**C**) RT-PCR of 16S rRNA from cell lysates (L) and the eluate (**E**) from the MS2 coat-protein column. (**D**) Scatter plot of ribosome occupancy (RO), the ratio of ribosome profiling to RNA-seq reads, from MS2RP of O-ribosomes vs. C-ribosomes. The red line indicates a 10-fold enrichment and the Pearson correlation is given. (**E**) Ribosome footprints (in reads per million mapped reads) from MS2RP of O-ribosomes and C-ribosomes on the *hemA* gene. The sequence upstream of the start codon is predicted to pair with the ASD of O-ribosomes.

The online version of this article includes the following figure supplement(s) for figure 1:

**Figure supplement 1.** Details of the expression of MS2-tagged mutant ribosomes.

**Figure supplement 2.** Enhanced translation of genes with ribosome-binding sites complementary to ASD-mutant ribosomes.

how well this purification strategy works: although signal from the wild-type 16S rRNA predominates in cell lysates (lower band, *Figure 1C*), it is nearly undetectable in purified ribosome samples eluted from the MS2-coat protein column. These data show that MS2-tagged ribosomes can be isolated with high purity for ribosome profiling studies; we refer to this procedure as MS2RP.

Comparison of the translational landscape of the canonical (C) to the orthogonal (O)-mutant confirms that the MS2RP strategy is effective. For 2217 genes with adequate coverage in each sample, we computed ribosome occupancy (RO) values by dividing the ribosome profiling density by RNA-seq density. Although we recognize that RO is not a perfect measure of initiation rates—it may also reflect differences in elongation in some cases—the number of ribosome footprints correlates strongly with protein levels in exponentially growing *E. coli* cultures (*Li et al., 2014*); RO therefore reports on the level of protein output per mRNA. We observed compelling differences in RO values for many genes in the two samples (*Figure 1D*). An initial straightforward expectation is that genes with SD motifs with high affinity to orthogonal (O) ASD sequence would have high RO values in MS2RP data from O-ribosomes; indeed, we observe that a complementary SD motif (UCCCG) five nt upstream of the start codon gives the *hemA* gene 10-fold higher RO with the O-ribosome than with the C-ribosome (*Figure 1E*). The same phenomenon was observed on *rbsK* (7-fold higher RO) and *mreB* (10-fold higher RO) with the O-ribosome and on *sapA* (9-fold higher RO) and *rsmH* (4-fold higher RO) with the S-ribosome (*Figure 1—figure supplement 2*). In each of these examples, the increase in RO can be attributed to higher levels of translation because the mRNA differs by less than two-fold. These examples are quite rare, however, because endogenous genes have evolved to interact with the canonical ASD and so the probability of finding a sequence with strong complementarity to the mutant ASD at just the right position is relatively low. Indeed, our data are most consistent with the conclusion that all three ASD mutants essentially act as general loss of function mutants.

## The global role of SD motifs on the endogenous translational landscape

We next used MS2RP to isolate the effect of SD motifs on global translation, asking to what extent they drive translation under optimal growth conditions. For each gene, we computed the SD strength as the inverse of the free energy (-ΔG) of pairing between the sequence −15 to −6 nt upstream of the start codon and the wild-type ASD (ACCUCCU). Based on the well-known role of SD motifs in promoting translational initiation, the expectation is that genes with strong affinity should have high RO values, and conversely, genes with weak affinity should have low RO values, yielding a strong correlation. However, our analysis of data from canonical (C) ribosomes showed only a very weak correlation (*Figure 2A*), consistent with previous reports from ribosome profiling studies (*Li et al., 2014*) showing that SD strength has little power to predict ribosome occupancy in *E. coli*. Strikingly, the RO values from the three ASD mutants (S, O, and A) showed a robust *negative* correlation with SD affinity for the wild-type ASD sequence (*Figure 2B* and *Figure 2—figure supplement 1*). In other words, ASD mutant ribosomes translate genes with weak SD motifs better than genes with strong SD motifs.

Because the ASD mutants are unlikely to participate in SD-ASD interactions, RO values in these samples reflect the contributions of all the other mRNA elements that promote initiation. The observation that these other elements yield a negative correlation with SD strength suggests that they in general counteract the positive correlation contributed by SD-ASD pairing (with wild-type ribosomes). As such, these contributions effectively mask the effect of SD motifs in *Figure 2A*. By calculating the difference in RO (ΔlogRO) for each gene between the C- and A-ribosomes, we effectively subtract all the mRNA elements that determine RO independent of SD-ASD pairing, thus isolating the effects of the SD motifs on mRNA translation rates. The ΔlogRO term reflects how much better a message is translated by wild-type ribosomes than by ASD mutants. When ΔlogRO values are plotted as a function of SD-ASD affinity (-ΔG) using the wild-type ASD sequence, we observe a strong linear correlation with SD-ASD affinity for each of the mutants (*Figure 2C* and *Figure 2—figure supplement 1*). As expected, genes with strong SD motifs are translated better by ribosomes with the canonical ASD than by ASD-mutant ribosomes. The fact that we observe this correlation validates our calculations of SD strength; analysis of the distance of SD motifs from the start codon confirms that genes with the highest ΔlogRO have the strongest SD affinity in the −15 to −6 region as shown in previous studies (*Figure 2—figure supplement 2*). These data obtained with MS2RP reveal for

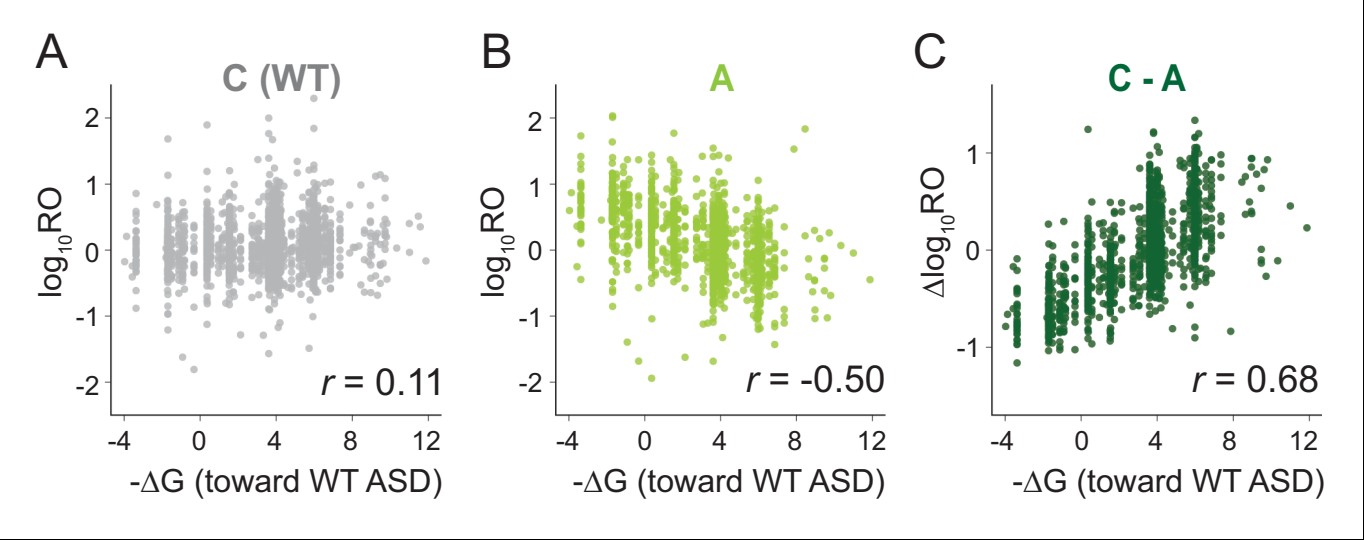

**Figure 2.** MS2RP reveals that SD motifs enhance translation genome-wide. Ribosome occupancy (RO) is the ratio of ribosome profiling to RNA-seq reads per gene. Log$_{10}$RO values are plotted against the SD strength (-ΔG of pairing to the wild-type ASD) for each gene with MS2RP data for C-ribosomes (A) and A-ribosomes (B). (C) Scatter plot of ΔlogRO (C-ribosomes minus A-ribosomes) and -ΔG where *r* values indicate Pearson correlations.

The online version of this article includes the following figure supplement(s) for figure 2:

**Figure supplement 1.** MS2RP reveals that SD motifs enhance ribosome occupancy.

**Figure supplement 2.** Genes translated better by wild-type ribosomes than by ASD mutants have sequences between −15 and −6 nt upstream of the start codon with high affinity to the ASD.

the first time the effect of SD motifs on translation genome-wide, consistent with their characterized role in promoting initiation.

## SD motifs are not necessary for start codon selection

SD motifs are also widely held to play a critical role in recognizing and selecting initiation sites (*Steitz and Jakes, 1975*). In the analyses described so far, we have used MS2RP to estimate the ribosome density on each mRNA as a proxy for initiation rates in order to address questions about *how much* translation is occurring on annotated genes. These data are less informative about the degree to which mutant ribosomes initiate at the wrong sites in the transcriptome. Non-canonical initiation is difficult to observe in *E. coli* because 5'- and 3'-untranslated regions of mRNAs are generally quite short and translation at alternate start codons within ORFs is swamped by the signal of elongating ribosomes from the canonical start site. In eukaryotes, the antibiotics harringtonine and lactimidomycin have been used with great success together with ribosome profiling to identify sites where translational initiation takes place (*Ingolia et al., 2011*; *Lee et al., 2012*). These compounds do not interfere with elongating ribosomes, allowing them to continue translation and terminate normally. In contrast, they trap newly-initiated ribosomes, providing a way of identifying initiation sites in ribosome profiling studies. Two antibiotics were recently shown to similarly specifically trap initiation complexes in bacteria: Onc112 and retapamulin (*Meydan et al., 2019*; *Weaver et al., 2019*).

To study the role of SD motifs on start codon selection, we treated cells with retapamulin for 5 min and then used MS2RP to identify start sites occupied by ribosomes with the various ASD sequences. For example, elongating wild-type (C) ribosomes are found all across the *lpp* gene in untreated cells (*Figure 3A*, light grey), whereas they are highly enriched at the annotated start codon in retapamulin-treated cells (dark grey). As expected, ribosome footprints are not seen at three internal AUG codons, since these do not function as initiation sites. Strikingly, in retapamulin-treated cells, the A-ribosomes also find the correct start site, ignoring the three other AUG codons (*Figure 3A*, dark green). In another example, the *gmk* gene, both C- and A-ribosomes are enriched at the annotated start codon in retapamulin-treated cells but not at several internal AUG codons

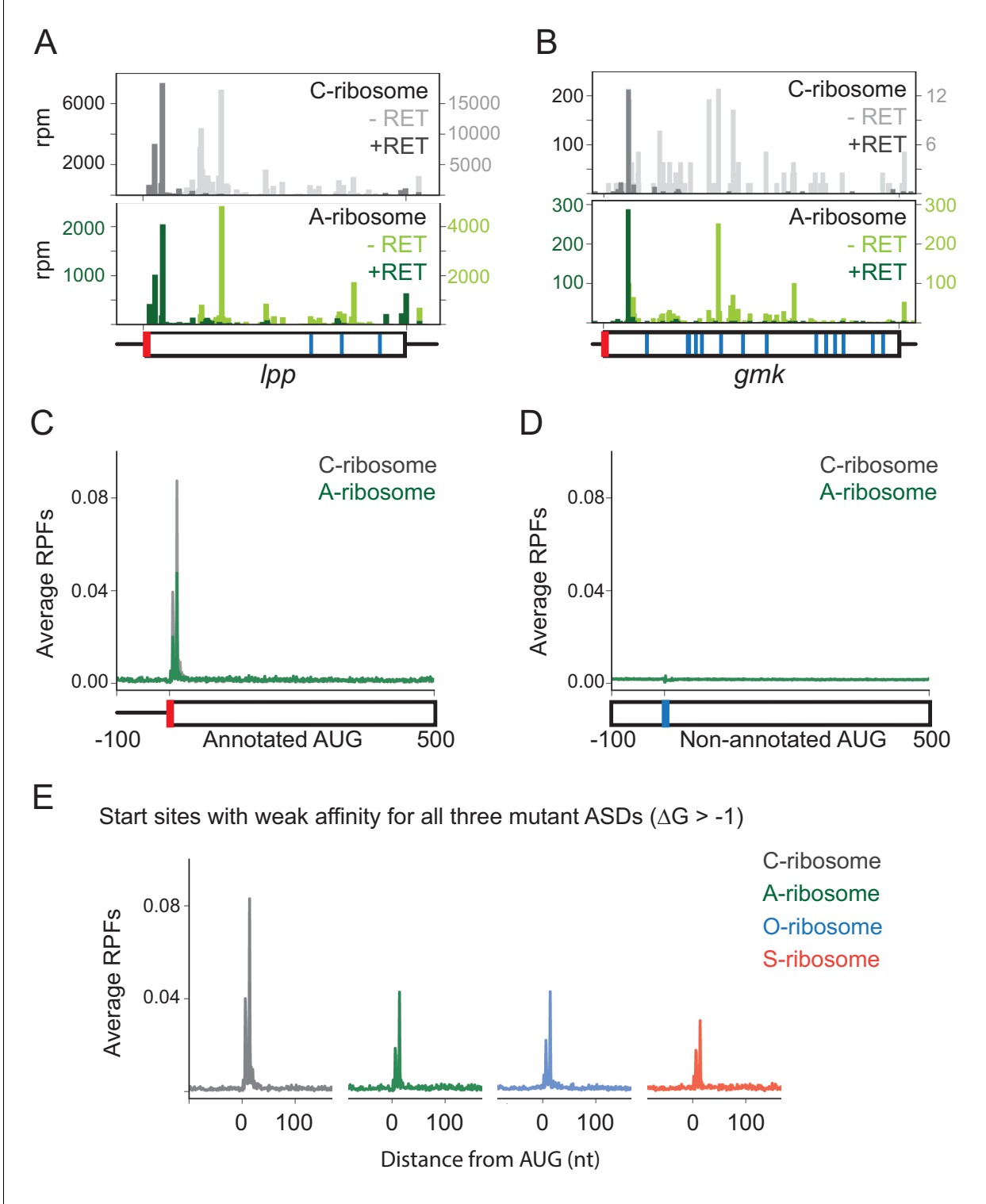

**Figure 3.** Loss of SD-ASD pairing has little effect on start codon selection. (A,B) Ribosome footprints on *lpp* and *gmk* from MS2RP data obtained with and without retapamulin, an antibiotic that traps ribosomes at start codons. Annotated AUGs are indicated by a red bar, non-annotated AUGs are indicated by blue bars. (C, D) Average ribosome protected fragments (RPFs) at annotated AUGs and non-annotated AUGs (where AUG starts at 1). (E) Average RPFs at the start codon of genes whose ribosome-binding sites have little or no affinity to all three mutant ASD sequences.

The online version of this article includes the following figure supplement(s) for figure 3:

**Figure supplement 1.** Ribosome density at annotated start sites does not depend on SD-ASD base pairing.

(*Figure 3B*). In both examples, both WT and mutant ribosomes select the correct, annotated start site while ignoring other AUG codons.

To analyze the accuracy of start codon selection by the ASD variants in retapamulin-treated samples genome-wide, we computed the average number of ribosome footprints across many genes aligned at their annotated start codons or aligned at all the other AUG triplets in the transcriptome (non-annotated AUGs). Our initial expectation was that in the absence of SD-ASD base pairing, the mutant ribosomes might fail to recognize the correct start sites and bind more often to other AUG triplets in the transcriptome. Strikingly, both the C- and A-ribosomes show strong initiation peaks at annotated AUGs (*Figure 3C*), whereas these peaks are absent in both samples at non-annotated AUGs (*Figure 3D*). These results provide initial evidence that ribosomes correctly select annotated start sites genome-wide in the absence of the SD-ASD interaction.

To further explore this surprising finding, we next asked how the affinity of mRNA-rRNA base pairing influences initiation at annotated start codons. We assumed that for the mutant ribosomes, base pairing would play little or no role in initiation because they would likely have low affinity for annotated start sites that evolved to bind the wild-type ASD. To test this assumption, we calculated the affinity of each mutant ASD for the sequence upstream of the start codon of each gene. We grouped genes into different sets based on these affinities and plotted the average number of ribosome footprints at the annotated start sites as in *Figure 3C*. In the subset of genes with no predicted affinity for any of the three ASD mutants ($\Delta G > -1$), we still see robust enrichment of A, O, and S ribosomes at the annotated start sites (*Figure 3E*). Since all three ASD variants initiate at annotated start sites, these data argue against the possibility that serendipitous base-pairing between the mRNA and the mutant ASD sequences explains this enrichment.

We also analyzed a set of annotated start sites with strong calculated affinity to the wild-type ASD. These sites are expected to be dependent on the SD-ASD interaction. Yet we again observed robust start peaks for each ASD variant ribosome, indicating that SD-ASD pairing is dispensable for initiation even in genes with strong SD motifs (*Figure 3—figure supplement 1A*). Furthermore, we found that in a set of sites with predicted high affinity to the ASD of the O-ribosome, there was strong enrichment of A- and S-ribosomes at start codons, despite the differences in the ASD sequence (*Figure 3—figure supplement 1B*). Likewise, in a set of genes with predicted high affinity to the ASD of the S-ribosome, there was strong enrichment of O- and A-ribosomes at start codons (*Figure 3—figure supplement 1C*). (There were too few genes with high affinity to the A-rich ASD sequence to perform an equivalent analysis for A-ribosomes). Taken together, these analyses show that annotated initiation sites are hard-wired for initiation independent of their potential for base pairing between the mRNA and rRNA.

## SD motifs are not necessary for initiation at non-canonical sites

We next asked what role mRNA-rRNA pairing plays in initiation at AUG triplets in the transcriptome that are not normally used for initiation (non-annotated AUGs). For this purpose, we used data from retapamulin-treated cells to calculate an initiation score (IS) for each AUG triplet, defined as the average number of reads mapped within 3 to 21 nt downstream of an AUG (to capture footprints of various sizes) divided by the average number of reads mapped over a wider spacing (100 nt, *Figure 4A*). The first and most general finding is that the $\log_2 IS$ values from the C- and A-ribosomes have a similar distribution with medians close to 0 (*Figure 4B*), indicating that footprints from the A-ribosomes are not enriched at non-annotated AUG codons. This result is consistent with the average gene plot shown in *Figure 3D* and with the fact that most of these AUG codons do not serve as initiation sites. To better characterize the difference between C- and A-ribosomes in initiation at non-annotated AUG codons, we selected a subset of sites that effectively recruit C-ribosomes and yield strong initiation peaks. These sites have $\log_2 IS$ values > 1.5 and are highlighted in black in *Figure 4B*. Surprisingly, this same subset of AUG codons also shows high IS values for A-ribosomes (*Figure 4C*), arguing that SD-ASD pairing is not the feature that explains why initiation takes place at these specific AUG triplets and not at others.

To further characterize how SD-ASD pairing affects initiation at non-annotated AUG triplets, we grouped potential initiation sites by their affinity for wild-type or mutant ASDs as described above for annotated start sites. For sites with high affinity to the ASD of the S-ribosome, for example, the distribution of IS values for S-ribosomes closely resembled the other three ribosomes (*Figure 4E*), with median values near zero. These data show that the presence of a complementary Shine-

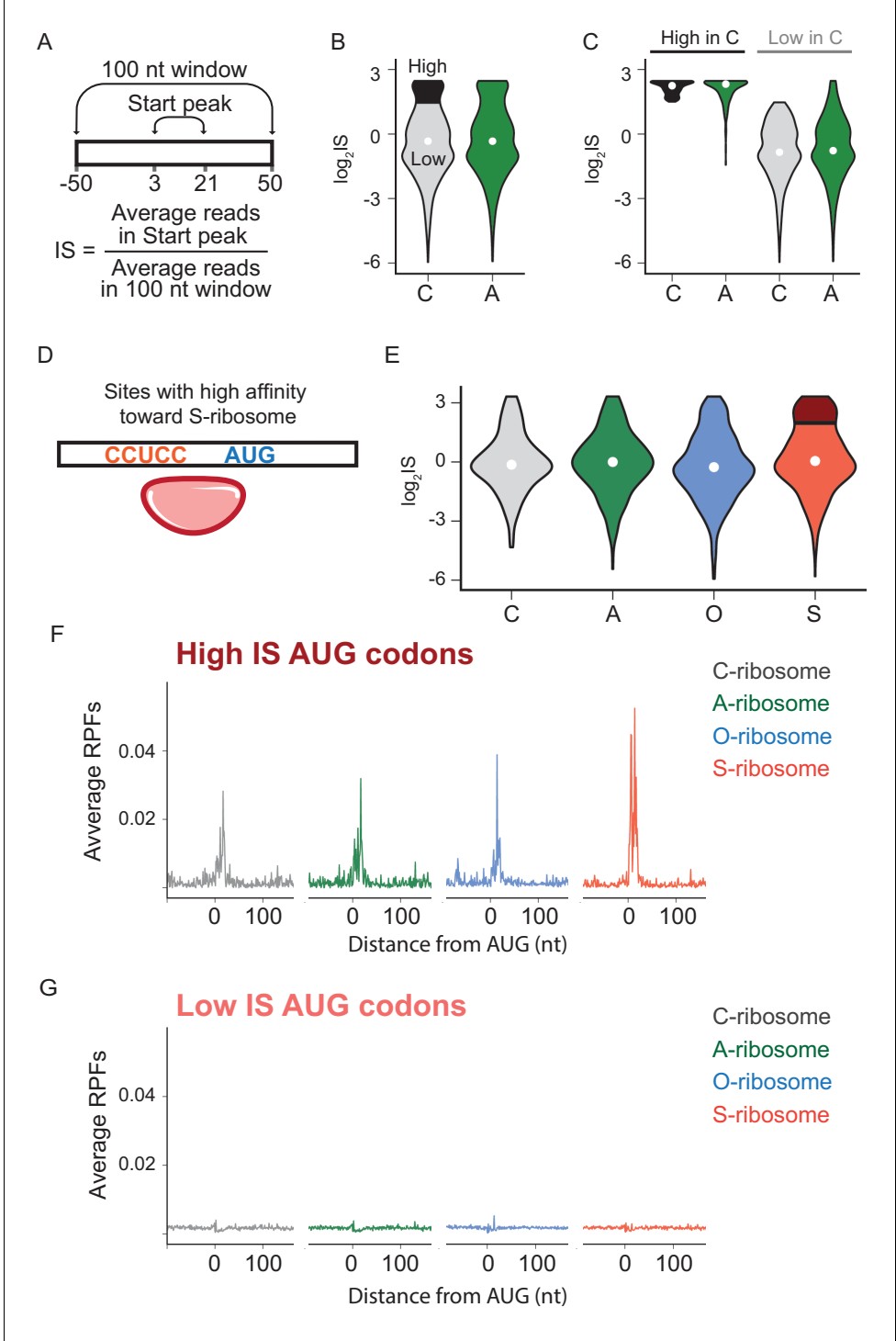

**Figure 4.** The effects of SD-ASD pairing on initiation at non-canonical sites. (A) Evaluation of initiation score, IS. (B) Initiation scores on non-annotated AUG triplets. For C-ribosomes, the fraction with IS >1.5 is colored black. (C) Initiation scores for C- and A-ribosomes for the set of sites with IS >1.5 for C-ribosomes (High, colored black in B) and those with IS <1.5 (Low). (D,E) IS values for all four ribosome types on the subset of sites with high affinity for the ASD of the S-ribosome (CCUCC). Average RPFs at the AUG triplets with high IS scores (F) or low IS scores (G) from the S-ribosome data. The online version of this article includes the following figure supplement(s) for figure 4:

**Figure supplement 1.** The effects of SD-ASD pairing on initiation at non-canonical sites.

Dalgarno-like sequence near an AUG codon is not sufficient to recruit S-ribosomes and generate a robust start codon peak. We selected the subset of AUGs with high affinity to S-ribosomes where initiation occurs with S-ribosomes ($\log_2$IS >1.5, dark red in **Figure 4E**). As expected, these high-IS sites show strong start peaks with S-ribosomes; however, the other ribosomes with different ASD sequences show robust start peaks as well (**Figure 4F**). Similarly, low-IS sites that are not translated by S-ribosomes (light red in **Figure 4E**) are also not translated by the other ribosomes (**Figure 4G**). The observation that SD-ASD pairing does not contribute to initiation at these sites with high affinity to the S-ribosomes also holds true for non-annotated AUGs with high affinity to the wild-type ASD (**Figure 4—figure supplement 1**). Once again, these data argue that AUGs that recruit ribosomes and lead to initiation are hard-wired for this purpose irrespective of the strength of the mRNA-rRNA base pairing interaction. Taken together, these data on initiating ribosomes show that mRNA-rRNA base pairing is neither necessary nor sufficient for translational initiation.

## A-rich sequences upstream of start codons promote initiation

To provide insight into mRNA features other than SD strength that might contribute to ribosome recruitment, we asked which features are enriched at annotated start sites. To avoid interference from SD motifs, we selected only annotated start sites with low affinity to the wild-type ASD ($\Delta G > 0$) and compared them to non-annotated AUG codons, most of which do not lead to initiation. We observed enrichment of adenosines (A) at many sites within 15 nt upstream of the start codon and 5 nt downstream (**Figure 5A**).

To test whether these A's promote translation, we selected four mRNAs with A-rich initiation sites (and weak SD motifs) and established a GFP reporter assay to follow their activity (**Figure 5B**). Of these four mRNAs (**Figure 5C**), two contain *annotated* initiation sites with low ASD-affinity, the start codons from *yhbY* and *gsk*. We also selected two representative *non-annotated* AUG codons found within the *creA* and *yeiR* genes; these sites have high IS values in both the C-ribosome and O-ribosome MS2RP data from retapamulin-treated cells. The sequences surrounding these four AUG codons (from 30 nt upstream to 45 nt downstream) were fused in frame to GFP such that GFP fluorescence reports on the activity of the AUG of interest. In addition to the wild-type sequence, we constructed mutants in which all of the A's 15 nucleotides upstream of AUG were changed to either U's or C's (G's were avoided because they have high affinity for the ASD). The reference protein mCherry was also expressed from the same plasmid with a standard ribosome binding site. The GFP/mCherry ratio was then normalized to a control lacking the GFP sequence (measuring only cellular auto-fluorescence).

We observed that the GFP/mCherry ratio was higher than background for all four AUG codons, showing that all are capable of driving GFP expression (**Figure 5D**). The two annotated start sites from *yhbY* and *gsk* induced stronger GFP expression than the non-annotated start sites, *creA\** and *yeiR\**. Importantly, however, the fact that fluorescence was observed from these latter examples confirms the results from the MS2RP data from retapamulin-treated cells showing that they are translated to some extent by wild-type ribosomes. We observed that replacement of the A's with U's lowered GFP expression in all cases except for *yeiR\** which showed the weakest GFP expression. A stronger effect was observed by changing the A's to C's, which led to complete loss of GFP fluorescence from all four AUG contexts tested. These results support our hypothesis that A-rich sequences upstream of start codons contribute to the identification of translational start sites.

The ability of A-rich sequences to promote initiation is likely not limited to *E. coli*: when we compared the local context of AUG codons in annotated start sites vs. non-annotated AUG codons for a set of diverse bacteria, we again saw that A-rich sequences were enriched (**Figure 5—figure supplement 1**). For *E. coli* and most other species examined, the enrichment of A's was weaker than the enrichment of G's corresponding to the SD sequence, but for *Mycoplasma pneumoniae* and *Flavobacterium johnsoniae*, the SD signal is not observed and there the enrichment of A's is particularly striking. A-rich sequences are highly conserved and may serve as an important mechanism for start site selection in these species, while contributing broadly to more diverse species.

## mRNA structure is lower at annotated start sites than at non-annotated AUG codons

In bacteria, mRNA structure surrounding the start codon has been shown in mechanistic studies to reduce ribosomal occupancy (*Lodish, 1970*; *de Smit and van Duin, 1990*; *de Smit and van Duin, 2003*; *Espah Borujeni and Salis, 2016*). Moreover, several transcriptome-wide analyses of mRNA structure in *E. coli* show lower levels of structure surrounding initiation sites (*Del Campo et al., 2015*; *Burkhardt et al., 2017*). We asked how mRNA structure differs between annotated start sites and internal AUG codons that are not annotated as start sites. We used data from a recent study of the structure of mRNAs in vivo using SHAPE and deep sequencing (*Mustoe et al., 2018*). From transcripts with sufficient coverage, we calculated the median SHAPE reactivity over a 120 nt window surrounding 365 annotated start sites and compared it to 7310 non-annotated AUGs (*Figure 5E*). For annotated initiation sites, the level of mRNA structure is significantly lower for a region 30 nt in length on both sides of the AUG codon (shown in red) as previously reported (*Del Campo et al., 2015*; *Burkhardt et al., 2017*). In contrast, except for a sharp dip in reactivity at the aligned AUG codon due to sequence bias, we see that mRNA structure is consistently high across this window for the set of non-annotated AUGs (shown in blue). These differences may be due in part to the ability of ribosomes to melt RNA structure during translation; indeed, initiation leads to the unfolding of RNA, which facilitates initiation by another 30S subunit (*Espah Borujeni and Salis, 2016*; *Andreeva et al., 2018*). But, given that SHAPE and DMS reactivity of mRNAs in vivo and in vitro are strongly correlated (*Burkhardt et al., 2017*; *Mustoe et al., 2018*), it is also likely that mRNA structure plays a causal role in setting initiation rates.

## Discussion

In this study, we performed ribosome profiling on mutant ribosomes purified using an RNA tag, the MS2 aptamer, a strategy we call MS2RP (*Figure 1*). Originally developed for in vitro studies of ribosomes containing lethal rRNA mutations (*Youngman et al., 2004*; *Youngman and Green, 2005*), MS2-tagged ribosomes also have potential to yield insights into the function of key rRNA sequences in vivo. In addition to the studies of the ASD sequence in 16S rRNA reported here, MS2RP could be employed to characterize the functions of rRNA domains on initiation, elongation, termination, and recycling at a genome-wide level in vivo. Because MS2RP can be performed on rRNA mutants expressed from plasmids, the method can be easily transferred to other bacteria or to eukaryotes without altering rDNA in the genome. Of particular interest are rRNA variants in bacterial genomes that are expressed differentially in response to changes in the environment and are proposed to have different specificities or functions (*Kurylo et al., 2018*; *Song et al., 2019*). Variant rRNA alleles have also been reported for eukaryotic cells (*Parks et al., 2018*); for example, different small subunit rRNA alleles are expressed in various developmental stages in Plasmodium (*Gunderson et al., 1987*). In addition, the functions of the highly variable rRNA expansion segments in eukaryotes are poorly understood (*Spahn et al., 2001*; *Anger et al., 2013*). MS2RP could be a powerful tool to elucidate the activities of various subpopulations of variant or mutated rRNAs.

Previous genome-wide studies in bacteria have shown little or no correlation between SD strength and ribosome occupancy (*Li et al., 2014*; *Schrader et al., 2014*; *Del Campo et al., 2015*). Using MS2RP, we are able for the first time to reveal the role of SD motifs in promoting initiation across the transcriptome. In our approach, we mutated the ASD on the ribosomes, thus maintaining mRNA sequence and structure, thus allowing us to isolate the effects of the SD:ASD interaction on translation. In the absence of SD:ASD pairing, we observed a strong negative correlation between ribosome occupancy and the SD strength (calculated by pairing with the wild-type ASD sequence). In other words, the mutant ribosomes translate genes with strong SD motifs worse than those with weak SD motifs (*Figure 2B*). There are two possible explanations for this negative correlation. It may be that the binding of wild-type ribosomes to mRNAs with strong SD motifs occludes their ribosome-binding sites, preventing mutant ribosomes from initiating and efficiently translating these genes. Alternatively, mRNA structure and other features may outweigh the impact of SD motifs, masking their effects, explaining why conventional ribosome profiling studies failed to observe correlations between SD strength and ribosome occupancy. Regardless of which of these explanations is correct, the MS2RP strategy allows us to subtract the cumulative contribution to ribosome occupancy of all of such other mRNA features, and thus to focus exclusively on the contribution to

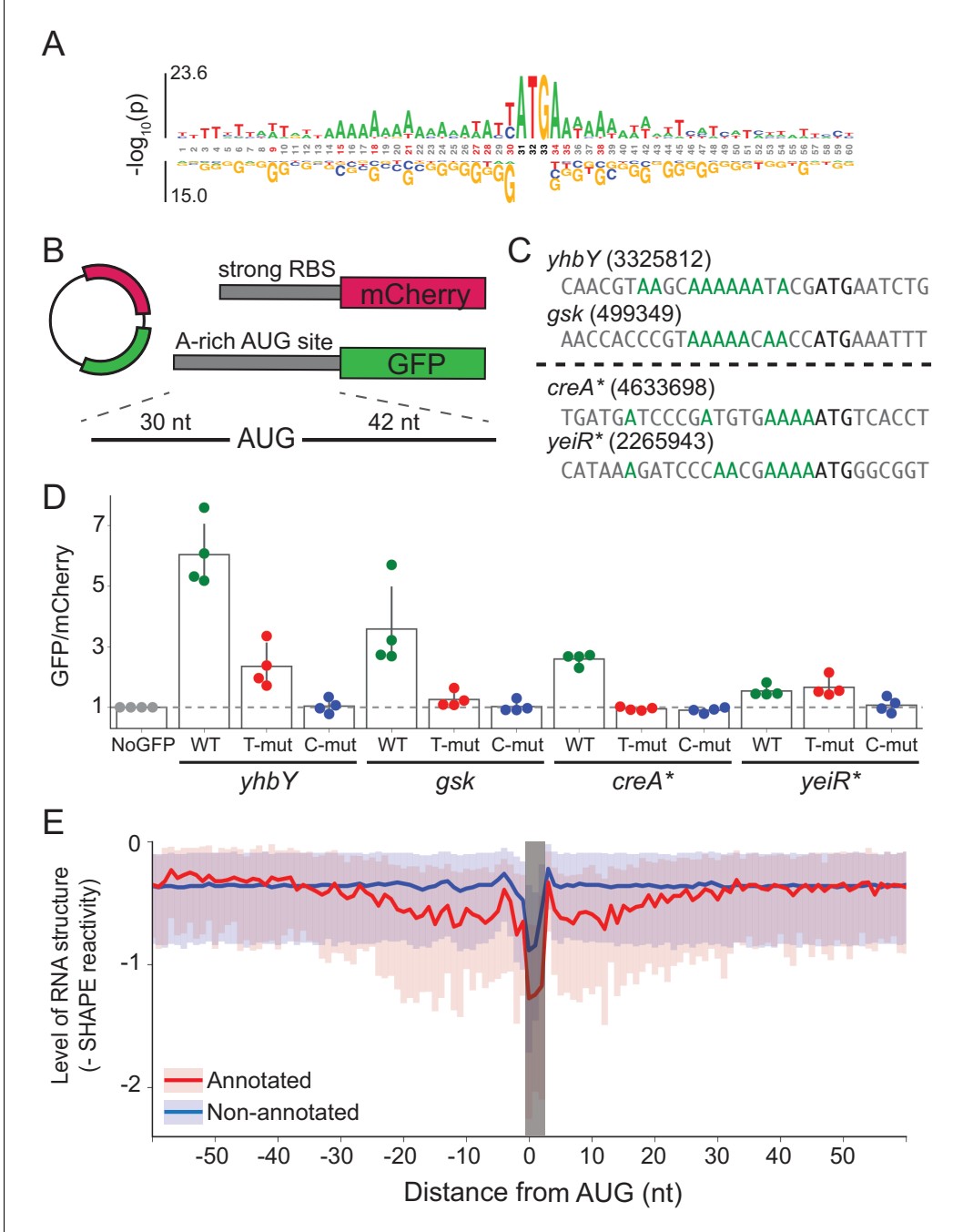

**Figure 5.** A-rich sequences as a signal for start codon selection. (**A**) Probability logo of the region surrounding annotated AUGs with low affinity to the wild-type ASD sequence (ΔG > 0) as compared with all non-annotated AUGs in the transcriptome. Enriched nucleotides are shown above the axis and depleted nucleotides below the axis. The height of the letter represents the binomial P-value. (**B**) Design of the reporter assay. The reporter plasmid encodes mCherry with a strong ribosome binding site (RBS) and separately GFP downstream of a region containing a start site of interest (30 nt upstream of AUG and 42 nt downstream). (**C**) Initiation sites used in the reporter assay; the number indicates the genomic position of AUG. In the T- and C-mutants, the A's upstream of AUG (highlighted in green) were substituted by T or C. (**D**) Results of the reporter assay. Each dot is the median of GFP/mCherry from an independent run of flow cytometry. The bar graph indicates the mean and SD from four independent tests. NoGFP (a plasmid that encodes mCherry but not GFP) serves as a control showing the baseline signal from cellular autofluorescence; the other data are normalized to this ratio. (**E**) Median (solid line) and interquartile range (shaded) of mRNA structure in SHAPE-MaPseq data for 365 annotated start sites (red) and 7310 non-annotated AUGs within coding sequences (blue).

The online version of this article includes the following figure supplement(s) for figure 5:

**Figure supplement 1.** A-rich sequences near start codons throughout the Bacterial kingdom.

ribosome occupancy of the SD:ASD interaction genome-wide. In this analysis, we are now able to see a linear correlation between the SD strength of an mRNA and protein output (*Figure 2C*).

Given that the SD motif functions through a well-defined mechanism and is widely conserved throughout bacteria, it has been thought to provide an important mechanism for start codon selection and translational output. Consistent with such a view, SD motifs are underrepresented within ORFs in order to avoid spurious initiation at internal start codons (*Hockenberry et al., 2018*). Strikingly, however, we find that ribosomes with altered ASDs still find the correct start codons about as efficiently as wild-type ribosomes (*Figure 3*). Start peaks for all four ribosome types are observed at annotated start sites regardless of the affinity of the ribosome binding site for the ASD. This shows that initiation sites are hard-wired for initiation based on mRNA features separate from the potential for SD-ASD pairing. These observations also hold true at the occasional non-annotated AUG codons where some initiation occurs (*Figure 4*). These data are consistent with the conclusion that SD motifs are not essential for determining where translation starts on mRNAs genome-wide.

What, then, are other mechanisms that could be used for start codon selection? Local mRNA structure and RNA folding kinetics clearly must play a critical role in allowing ribosomes to find the start codon. A number of mechanistic studies have demonstrated that RNA structure around the start codon lowers translation levels (*Hall et al., 1982*; *de Smit and van Duin, 1990*; *Osterman et al., 2013*; *Espah Borujeni et al., 2014*). Studies of factors that alter the expression of simplified reporter genes (involving randomization of the 5'-UTR or coding sequences) show that lack of secondary structure surrounding the initiation site has the most significant correlation with protein output (*Salis et al., 2009*; *Kudla et al., 2009*; *Goodman et al., 2013*). Recent transcriptome-wide analyses of mRNA structure in *E. coli* confirm that annotated start sites have lower levels of mRNA structure, as seen by PARS on purified mRNA and DMS-seq in vivo (*Del Campo et al., 2015*; *Burkhardt et al., 2017*). mRNA structure is likely an important factor in start site selection: using high-resolution SHAPE-MaPseq data (*Mustoe et al., 2018*), we showed that annotated AUGs have lower levels of RNA structure 30 nt upstream and downstream whereas internal AUG are not surrounded by regions of lower structure (*Figure 5E*).

Interestingly, in comparing the sequence context of AUG codons that are annotated as initiation sites with those that are not, we found that A's are enriched both upstream and downstream of annotated initiation sites (*Figure 5A*) and we confirmed their importance in reporter assays (*Figure 5B–D*). These results from endogenous initiation sites are reminiscent of observations of the over-representation of A's in 5'-UTR sequences selected for strong affinity to the ribosome in vitro (*Gao et al., 2016*) and in 5'-UTRs selected from random sequences upstream of a reporter gene for high levels of translation in vivo (*Evfratov et al., 2017*). Comparison of annotated start sites and non-annotated AUGs across several bacterial genomes shows that this mechanism is widespread (*Figure 5—figure supplement 1*). Although enrichment of A's is more subtle than enrichment of G's in *E. coli* and *B. subtilis*, in organisms that lack SD motifs, such as *Mycoplasma pneumoniae* and *Flavobacterium johnsoniae*, A-rich motifs may play an important role in initiation. Indeed, in a recent study, Fredrick and co-workers used ribosome profiling in *F. johnsoniae* and observed enrichment of A's upstream of start codons in mRNAs with high ribosome occupancy in comparison to genes with are translated less efficiently (*Baez et al., 2019*). We envision that this sequence, like the Shine-Dalgarno motif, acts as a translational enhancer, fine-tuning the efficiency of initiation.

The mechanism by which A-rich sequences enhance initiation is not clear. The prevalence of A's may alter the mRNA dynamics; A-rich sequences tend to have less secondary structure than GC-rich sequences. We note however that replacing A's with U's in several reporters reduced translation levels even though the U's are similarly not expected to yield strong structures. A second possibility is that ribosomal components may interact specifically with A's close to the start codon that are bound inside the ribosome during initiation. Fredrick and co-workers used reporter assays to show that mutation of a particular A at the −3 position reduces expression; this result is intriguing because the classic Kozak sequence (GCC(A/G)CCAUG) that promotes high levels of translation in eukaryotes also contains a purine at position −3. A-rich sequences have been reported to enhance translation in a variety of eukaryotic contexts including *Drosophila* and wheat germ and reticulocyte lysates (*Ranjan and Hasnain, 1995*; *Sano et al., 2002*; *Suzuki et al., 2006*; *Pfeiffer et al., 2012*). It may be that A-rich sequences interact with conserved elements of the ribosome across the domains of life. A's further from the start codon (10–20 nt upstream) may interact with bacteria-specific ribosomal protein S1. bS1 preferably binds to A/U-rich sequence elements upstream of SD sequences

(*Boni et al., 1991*; *Komarova et al., 2005*) and is thought to unwind mRNA structure to induce initiation (*Qu et al., 2012*; *Duval et al., 2013*).

Our findings have broad implications for the evolution of translational mechanisms in bacteria. Not all bacteria utilize SD motifs to promote translational initiation—SD motifs are notably lacking in Bacteroidetes and Cyanobacteria. Because the prevalence of SD motifs is a feature of the genome in general and not of a single gene, it makes sense that evolutionary selective pressure for or against SD usage would act at the level of the transcriptome. The nature of these selective pressures remains unclear, although Hockenberry recently argued that bacteria with high levels of SD usage tend to have higher maximal growth rates (*Hockenberry et al., 2017*). Future studies will clarify the evolutionary relationship between the growth environment, levels of SD usage among bacterial species, and their transcriptome-wide effects.

## Materials and methods

### Growth conditions

Unless otherwise specified, cells were cultured at 37°C in 500 mL of LB + ampicillin (50 mg/L). IPTG was added (0.3 mM final) when the culture reached $OD_{600} = 0.3$ and cells were harvested by filtration at $OD_{600} = 0.5$. For profiling with retapamulin, cells were grown at 37°C in 500 mL of LB + ampicillin to $OD_{600} = 0.3$, induced with IPTG, grown to $OD_{600} = 0.45$, and then harvested by filtration 5 min after the addition of retapamulin (100 µg/mL final).

### Cell harvest and lysis

Cells were harvested by filtration using a Kontes 99 mm filtration apparatus and 0.45 um nitrocellulose filter (Whatman) and then flash frozen in liquid nitrogen. Cells were lysed in lysis buffer (20 mM Tris pH 8.0, 10 mM $MgCl_2$, 100 mM $NH_4Cl$, 5 mM $CaCl_2$, 100 U/mL DNase I, and 1 mM chloramphenicol) using a Spex 6870 freezer mill with 5 cycles of 1 min grinding at 5 Hz and 1 min cooling. Lysates were centrifuged at 20,000 g for 30 min at 4°C to pellet cell debris.

### Overexpression and purification of MBP-MS2-His protein

BL21(DE3) cells were transformed with the plasmid pMal-c2G-MBP-MS2-His, cultured at 37°C in LB + ampicillin (50 mg/L) to $OD_{600} = 0.7$, and induced with 0.3 mM final IPTG for 4 hr at 37°C. Cells were harvested by centrifugation and lysed on a french press in the binding buffer (50 mM $NaH_2PO_4$ pH 8.0, 300 mM NaCl, 10 mM imidazole, 6 mM BME). The MBP-MS2 protein was purified by FPLC (Atka, GE); after washes with the binding buffer, it was elution with the binding buffer supplemented with 200 mM imidazole.

### Affinity purification of MS2-tagged ribosomes

3 mL of amylose resin (NEB) were transferred to a Poly-Prep Chromatography Column (Bio-Rad) and washed 3 times with 10 mL of lysis buffer. 2.5 mg of MBP-MS2-His protein were loaded onto the amylose resin, incubated at 4°C for 1 hr, and washed twice with 10 mL of lysis buffer. For MS2RP, 1.5 mL of cell lysate and 15 µL of RNase T1 (1000 U/µL, Thermo) were loaded onto the MBP-MS2 resin, incubated at 4°C for 2 hr, and washed 3 times with 10 mL of lysis buffer. The resin was re-suspended in 1 mL of lysis buffer and 360 µg MNase was added to digest mRNA and remove the MS2 hairpin in rRNA, releasing the ribosomes from the column. Following a 2 hr incubation at 25°C, the flow-through was collected. Another 2 mL of lysis buffer was passed through the resin and collected. The flow-through fractions were then combined.

### Sucrose density gradient centrifugation

10–54% sucrose density gradients were prepared using the Gradient Master 108 (Biocomp) in the gradient buffer (20 mM Tris pH 8.0, 10 mM $MgCl_2$, 100 mM $NH_4Cl$, 2 mM DTT). 5–20 AU of *E. coli* lysate was loaded on top of sucrose gradient and centrifuged in a SW41 rotor at 35,000 rpm for 2.5 hr at 4°C. Fractionation was performed on a Piston Gradient Fractionator (Biocomp).

## Library preparation

Libraries for MS2RP and standard ribosome profiling are prepared as in *Woolstenhulme et al. (2015)* and *Mohammad et al. (2016)*. At least two biological replicates were performed for each MS2RP library as detailed in the GEO database entry. RNA-seq libraries were prepared with TruSeq Stranded Total RNA Gold from 250 ng of total RNA following depletion of rRNA by RiboZero rRNA Removal Kit for bacteria (Illumina). Libraries were analyzed by BioAnalyzer high sensitivity DNA kit (Agilent) then sequenced on the HiSeq2500 (Illumina).

## Analysis of rRNA purity by RT-PCR

RNA was purified by hot-phenol extraction. The first strand synthesis was performed with 500 ng of total RNA, primer MS2check_R (5'-AGACATTACTCACCCGTCCGCCACTC-3') and SuperScript III (Invitrogen). 15 cycles of PCR amplification were performed with primer MS2check_F70 (5'-TGCAAG TCGAACGGTAACAGGAAG-3'), primer MS2check_R, and Phusion polymerase (NEB). PCR products were resolved by 8% TEB gel and analyzed by Typhoon FLA 9500 (GE).

## GFP/mCherry assay

MG1655 cells carrying the reporter plasmid were cultured in LB + ampicillin (50 mg/L) to early log phase. Cells were diluted 50-fold in TBS. GFP and mCherry fluorescence were measured on a Guava easyCyte flow cytometer (Millipore Sigma).

## General processing of sequencing data

For libraries prepared by linker with UMI (rAppNNNNNNCACTCGGGCACCAAGGAC), perfectly matching reads (including 5'-end and 3'-end UMI) were converted to a single read by Tally (*Davis et al., 2013*). 3'-linker sequences were removed by Skewer (*Jiang et al., 2014*). The 5' end UMI added by the RT primer were removed by seqtk. Reads were aligned using bowtie version 1.1.2 (*Langmead et al., 2009*), first to the tRNAs, rRNAs, and the *ssrA*, *ssrS*, *lacI* and *ffs* genes. Reads that failed to align to those sequences were aligned to *E. coli* MG1655 NC_000913.2. Ribosome position was assigned by the 3'-end of aligned reads. RNA-seq data were assigned by the 5'-end of aligned reads.

## Calculation of ΔG

The affinity (ΔG) of the ASD and the sequence of a start codon was calculated for each mRNA using free_scan with '-l 0 –b 0' option to disallow internal loop and internal bulge (*Nakagawa et al., 2010*). The input sequences were −15 and −6 nt upstream of AUG and the reverse sequence of wild-type ASD (UCCUCCA) or the mutant ASD where appropriate.

## Analyses of genome-wide mRNA structural data

Average SHAPE reactivity was based on the SHAPE-MaP data (*Mustoe et al., 2018*). A median of the SHAPE reactivity from the region −25 to +25 upstream and downstream of the start codon was used as degree on RNA structure.

## Analyses of initiation peaks in samples treated with retapamulin

AUG codons were only included in the analysis of average ribosome density and initiation scores if they had more than 10 mapped reads in the window of −50 upstream and +50 downstream of the AUG. To calculate average ribosome density, for each AUG we took the rpm at each position across this window, divided it by the total rpm in the window, and then computed the mean of these values for all AUGs included in the calculation. Initiation scores were computed by taking the mean of reads mapped within +3 to +21 nt downstream of the A in AUG and dividing it by the mean of reads mapped on the region −50 to +50 of the AUG.

## Probability logo

Probability logos were generated by kpLogo (*Wu and Bartel, 2017*) using its default settings. For *Figure 5A*, input and background sequences are described in the figure legend. For *Figure 5—figure supplement 1* the set of input sequences consisted of annotated AUGs from the GFF file

available at NCBI and the set of background sequences consisted of all AUGs in the genome that were not annotated as initiation sites.

## Data availability

The sequencing data are available in processed WIG format at the GEO using accession number GSE135906 and as the raw FASTQ files at the SRA. Custom python scripts used to analyze the sequencing data are freely available at https://github.com/greenlabjhmi/2019_SDASD (*Saito, 2020*; copy archived at https://github.com/elifesciences-publications/2019_SDASD).

# Acknowledgements

The authors thank Daniel Goldman, Colin Wu, and Boris Zinshteyn for critical reading of the manuscript, as well as David Mohr at the Genetics Resources Core Facility, Johns Hopkins Institute of Genetic Medicine, for sequencing assistance. This study was funded by a JSPS fellowship (KS), NIH grant GM110113 (ARB), and HHMI (RG).

# Additional information

### Competing interests

Rachel Green: Reviewing editor, *eLife*. The other authors declare that no competing interests exist.

### Funding

| Funder | Grant reference number | Author |
|---|---|---|
| National Institute of General Medical Sciences | GM110113 | Allen R Buskirk |
| Howard Hughes Medical Institute | | Rachel Green |
| Japan Society for the Promotion of Science | | Kazuki Saito |

The funders had no role in study design, data collection and interpretation, or the decision to submit the work for publication.

### Author contributions

Kazuki Saito, Conceptualization, Formal analysis, Investigation, Methodology, Writing - original draft; Rachel Green, Conceptualization, Funding acquisition, Writing - review and editing; Allen R Buskirk, Conceptualization, Formal analysis, Supervision, Funding acquisition, Writing - review and editing

### Author ORCIDs

Rachel Green [iD] http://orcid.org/0000-0001-9337-2003
Allen R Buskirk [iD] https://orcid.org/0000-0003-2720-6896

### Decision letter and Author response

Decision letter https://doi.org/10.7554/eLife.55002.sa1
Author response https://doi.org/10.7554/eLife.55002.sa2

# Additional files

### Supplementary files

• Transparent reporting form

### Data availability

Sequencing data have been deposited in the GEO under accession code GSE135906.

The following dataset was generated:

| Author(s) | Year | Dataset title | Dataset URL | Database and Identifier |
|-----------|------|---------------|-------------|-------------------------|
| Saito K, Green R, Buskirk AR | 2019 | Translational initiation in *E. coli* occurs at the correct sites genome-wide in the absence of mRNA-rRNA base-pairing | https://www.ncbi.nlm.nih.gov/geo/query/acc.cgi?acc=GSE135906 | NCBI Gene Expression Omnibus, GSE135906 |

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
