## [Decision Letter]

**Acceptance summary:**

This paper uses an innovative twist on ribosome profiling to investigate the importance of Shine-Dalgarno sequences in bacterial translation initiation. Surprisingly, the data show that strong base-pairing between the 16S ribosomal RNA and the mRNA Shine-Dalgarno sequence is neither necessary nor sufficient for translation initiation. This suggests that start-codons are "hard-wired" into the genome, largely independent of Shine-Dalgarno sequence.

**Decision letter after peer review:**

[Editors’ note: the authors submitted for reconsideration following the decision after peer review. What follows is the decision letter after the first round of review.]

Thank you for submitting your work entitled "Shine-Dalgarno sequences fine-tune translation genome-wide but are not the primary determinants of start-site selection" for consideration by *eLife*. Your article has been reviewed by three peer reviewers, including Joe Wade as the Reviewing Editor and Reviewer #1, and the evaluation has been overseen by Jim Manley as the Senior Editor. The following individual involved in review of your submission has agreed to reveal their identity: Shura Mankin (Reviewer #2).

Our decision has been reached after an extensive discussion involving the three reviewers. The reviewers were enthusiastic about parts of the manuscript, in particular the method itself; however, there was some disagreement as to the significance of the work. Much of the discussion focused on the data in Figure 6, which the reviewers considered to present potentially the most important result. We felt that further analysis is needed to fully address the key questions of (i) whether annotated start codons are inherently good at binding ribosomes, independent of the SD, and (ii) whether an SD alone (e.g. next to an ATG in the middle of an ORF) is insufficient to bind a ribosome. Put more simply, we felt that further analysis is required to show that start-codons are "hard-wired" into the genome, largely independent of SD sequence. We are therefore rejecting the paper because the outcome of the new analysis is unclear. Nonetheless, we would be willing to consider a revised version if the analyses suggested below, or something equivalent, provide stronger support for the idea that start-codons are hard-wired, independent of the SD sequence.

The main concerns with Figure 6 are that (i) the control set of ORF-internal ATGs is not the best control because many (most?) of the ATGs don't have a good SD sequence for the modified ribosomes; and (ii) ribosome density at annotated start codons for the modified ribosomes could be due to a subset of start codons that have decent matches to the modified ASD sequence. We suggest that a more appropriate control set of ATGs would be those with a good predicted match to the altered ASD sequence. We also suggest limiting the analysis in Figure 6C to start codons that have a poor match to the modified ASD. Another way to look at this would be to compare which annotated start codons are recognized by the different modified ribosomes; if all three types of ribosome recognize the same subset of start codons, it's safe to conclude that this is occurring independent of the SD. If these (or other) analyses can provide stronger support for the "hard-wired" model, that would likely be sufficient for publication. In addition to the re-analysis of data from Figure 6, it's important to improve the clarity of the paper, which was at times confusing (see the detailed reviews for more information on this). Additionally, reviewer #3 makes some important points about the calculation of hybridization energies, such as considering a full, 9 nt SD sequence with variable spacing. Lastly, the manuscript would benefit from a clearer description of what is already known about features other than SD sequence that contribute to translation initiation (see comments from reviewer 3).

*Reviewer #1:*

This paper describes an innovative approach to probe the importance of Shine-Dalgarno (S-D) sequences in translation initiation in *Escherichia coli*. By performing ribosome profiling on modified ribosomes, the authors are able to observe translation by ribosomes with altered anti-S-D sequences. This method reveals that despite no correlation between S-D strength and translation levels, there is a contribution of S-D strength that is apparent when all confounding factors have been controlled for. Interestingly, this effect of S-Ds is lost during other growth conditions, although for cold shock that is largely consistent with previous work, and it is unclear what the mechanism is in stationary phase. While I think the topic is interesting and the primary method is ingenious, I'm not convinced that the authors have learned much about the relative importance of S-Ds in translation initiation. As they acknowledge, previous studies have failed to see a correlation between S-D strength and translation initiation levels, and the importance of secondary structure and of A-rich sequences has been described previously. The fact that predictions of S-D strength correlate with translation initiation levels once factors other than S-D have been accounted for indicates that these predictions are fairly accurate. This is important, since it accounts for the possibility that the lack of correlation between predicted S-D strength and translation initiation is because of our inability to predict S-D strength. However, the impact of this advance is small. I also have concerns about the interpretation of Figures 6 and 7 that impact the overall conclusions.

- The presentation of the cold shock data is confusing. The overall conclusion is that S-D-dependence is lost at almost all genes during cold shock. However, a small subset of genes appears to depend strongly on the S-D. The distinction between the effect on the majority of genes and the effect on a small subset should be explained more clearly. The simplest interpretation of these data is that most start codons are highly structured during cold shock, but those that are do not rely on their S-Ds. This model is largely consistent with previous work.

- I disagree with the interpretation of Figure 6. The data show that for the altered ribosomes, annotated start codons are used far more efficiently than the collection of all other ATG sequences within ORFs. However, there are many more ATGs within ORFs than annotated start codons, and even if translation relies heavily on S-D sequences, you would expect that most ATGs within ORFs would not be selected by alternative ribosomes because only a small subset will have appropriate S-D sequences, and many may be weakly expressed. My interpretation of these data is that alternative ribosomes do use annotated start codons, but there is no way to tell how selectively they do this. A more appropriate comparison would be of (i) annotated start codons to (ii) ATGs within ORFs where the ATG is associated with a sequence that is predicted to function as a good S-D for the alternative ribosome.

- Another concern I have with Figure 6 is that presumably some, and perhaps many of the annotated start codons will have good SD matches for the alternative ribosomes. Figure 2C suggests that the number with good matches will be fairly high. Is the ribosome density at annotated start codons simply due to the subset of start codons that have reasonable SD matches to the altered ASD? Another way to think about this is to ask whether the start codons contributing to the signal in Figure 6C are the same start codons that contribute to the signal in Supplementary Figure 5A-B.

- Figure 7E shows the importance of an A-rich sequence in the context of start codons lacking a good S-D. Similar to Figure 6, these data highlight the contribution of non-SD sequences to translation initiation, but they do not provide any information about the relative importance of the different sequence elements.

*Reviewer #2:*

Major findings:

The paper of Saito and al. examines the contribution of Shine-Dalgarno sequence (SD) to the translation efficiency in bacteria. Using a clever approach, the authors use ribosome profiling to compare mRNA occupancy by wt ribosomes and ribosomes with the altered anti-SD sequence (ASD). In confirmation of previous findings from the Weissman lab, they find that the general translation efficiency does not correlate with the predicted strength of SD-ASD interactions. However, when all the other factors are masked, they observe a strong dependence of the initiation rate on the strength of SD-ASD pairing. They also noted that a subset of genes expressed in the stressed cells depend heavily on recognition of the SD sequence by the ribosomes. One of the unexpected, but highly important findings is the observation that the ribosomes with the altered ASD can nevertheless correctly and selectively initiate translation at the known start sites underscoring the importance of factors other than SD-ASD interactions in the start codon selection. Importantly, the reported work reveals the prevalence of A-rich motifs in the ribosome binding sites of the genes with weak SD sequences in *E. coli* and other bacteria. This trend becomes especially prominent in the bacterial species that do not rely on SD-ASD interactions for translation initiation.

Critique:

This is an interesting, intriguing and important study. The results are nice and clean and the implications are important for unraveling the fundamental mechanism of translation initiation in bacteria. Although the paper is generally well written, it was hard at times to follow the authors logic and I strongly encourage the authors to try to clarify the message, which often was hard to extract.

1) Here are several examples:

- Abstract: The statement "We reveal a genome-wide correlation between the SD strength and translational efficiency" is followed by "this global correlation is lost and a subset of genes […] becomes [dependent] on SD motifs for translation". This is hard to digest.

- Figure 4C legend ("the strength of the SD motifs determines whether wild-type or ASD mutants are recruited to messages") is supposed to contrast Figure 4F legend ("the unstructured SD motifs can recruit wild type ribosomes more effectively than they recruit ASD mutants"). However, they sound nearly identical and thus, do not accurately communicate the point the authors apparently are trying to make.

- "genes with strong SD motif are translated better by ribosomes with canonical ASD": better in comparison with the ASD-mutant ribosomes or better in comparison with the genes with weak SD?

2) Aleksashin et al., 2019, have shown that altering ASD in 16S rRNA compromises rRNA maturation. Although the presence of unprocessed sequences at the 5' and 3' end of the ASD-mutant 16S rRNA would not likely change the general conclusions of the paper, hypothetically it could affect the functionality and elongation rate of the mutant ribosomes. I am wondering whether authors have checked how well their mutant 16S rRNAs are processed. Irrespectively, I believe a more detailed discussion of the general functionality of the ribosomes with altered ASD, especially in relation to the elongation rate, would be beneficial.

3) Subsection “Gene-specific roles of SD motifs under stress”. The readers need a better explanation why the authors switched from ΔlogTE to ΔlogRPKM metrics when they move to the experiments in the stressed cells.

4) The influence of the competition between wt and mutant 30S subunits for the translation start sites on the conclusions drawn from ribosome profiling should be discussed.

*Reviewer #3:*

The authors introduce mutated 16S ribosomal RNAs into *E. coli* strains, altering their anti-Shine Dalgarno sequences, to investigate how these perturbations affect translation rates across the *E. coli* genome. To do this, they carry out ribosome profiling experiments to measure genome-wide ribosome densities on aSD-modified strains, including during exponential, stationary, and cold shock growth phases. Overall, they find that changing the last 9 nucleotides of the 16S rRNA has a significant effect on the transcriptomes' translation rates.

Overall, the collected measurements are interesting and potentially useful. However, the analysis suffers from a terribly incomplete knowledge of what controls a mRNA's translation rate. The statistics applied are tailored for a 1-factor problem, when in fact, there are many factors that control translation rate. There are also inconsistencies and errors in the authors' calculations that should be corrected. The authors' conclusions are not well supported by their analysis. The manuscript requires significant work for it to productively add to our knowledge of what controls translation rate in bacteria.

1) The author focuses primarily on the importance of the sequence colloquially known as the Shine-Dalgarno in controlling a mRNA's translation initiation rate. The authors write that "Initiation rates vary depending on how well an mRNA recruits 30S subunits to the start codon, and in bacteria, the working model is that this is accomplished primarily by Shine-Dalgarno (SD) motifs." This is incorrect. The current working model is that a mRNA's translation initiation rate is controlled by at least five important molecular interactions, only one is the hybridization between the last 9 nucleotides of the 16S rRNA and the mRNA. They include:

a) the hybridization between the last 9 nucleotides of the 16S rRNA and the mRNA;b) the unfolding of mRNA structures that overlap with the ribosome footprint;c) the differences in spacing (physical distance) between the 16S rRNA binding site and the start codon;d) the standby site's accessibility, as determined by the length of available single-stranded RNA;e) the start codon and its hybridization to the tRNA.

The free energy needed to unfold inhibitory mRNA structures is also affected by the dynamics of RNA folding (RNA folding kinetics) as well as the rate of ribosome binding.

If the authors better understood how translation rate was controlled, they could more productively use their measurements to push the real state-of-the-art forward. Their current conclusions are already subsumed within the state-of-the-art (i.e., nothing new).

2) Any discussion of "which translation rate interaction is most important" or "which translation rate interaction is responsible for X whereas the interaction Y only fine-tunes Z" is not productive and can easily be contradicted by selecting a real counter example. Overall, it is the binding free energy of the 30S ribosome to the mRNA that determines its translation initiation rate. Each of these interactions contributes free energy to this process and the magnitude of the contributed free energies can be roughly equal across a selection of real mRNA examples. There are unstructured mRNAs where there is little penalty for unfolding inhibitory mRNA structures. There are highly structured mRNAs that have consensus SDs sequences. There are mRNAs that have consensus SD sequences far away from the start codon. All of these mRNAs could have the same translation rate. Which interaction is most important? That's not the right question to ask, because it's meaningless.

3) The manuscript's main topic is the Shine-Dalgarno sequence, but the authors should be made aware that at least the last 9 nucleotides of the 16S rRNA can contact the mRNA and hybridize to it. In *E. coli*, the anti-Shine Dalgarno sequence is 5'-ACCUCCUUA-3' and the "consensus" Shine-Dalgarno sequence is therefore 5'-TAAGGAGGT-3'. The manuscript text and the authors' calculations should reflect this.

4) The authors are mis-using the ribosome profiling measurements in their analysis. Ribosome profiling measurements do not directly measure translation rates. They measure mRNA-bound ribosome densities. A mRNA's ribosome density will depend on both its translation initiation rate AND its translation elongation rate. Specifically, in steady-state conditions, the ribosome density will be the ratio between these two quantities (initiation rate over elongation rate). In the initial applications of ribosome profiling, researchers assumed that all mRNAs have the same translation elongation rate in order to conclude that ribosome density measurements were proportional to translation initiation rates. This is not true. Coding sequences in mRNAs have very different translation elongation rates, due to differences in synonymous codon usage. Unless each mRNAs' translation elongation rates are predicted or directly measured, ribosome density measurements cannot be used to infer their translation initiation rates. Therefore, when the authors write "In pioneering ribosome profiling studies in bacteria, the paradoxical observation was made that there is little or no correlation between the translational efficiency of a gene and the strength of its SD motif (calculated using thermodynamic algorithms for RNA pairing), as had been anticipated based on the SD model." there is no actual paradox. The ribosome profiling measurements were not used correctly to test how mRNA sequences control translation rate.

5) Getting to the authors' main conclusions, they write that "These data indicate that the ASD mutant ribosomes translate genes with weak SD motifs better than genes with strong SD motifs, exactly the opposite of what wild-type ribosomes are expected to do." This statement is confusing given the real conclusion of the authors, that all other factors being equal a "strong" SD motif does result in higher translation than a "weak" SD motif. It's only because of other confounding factors that the initial analysis did not yield a positive correlation. An incorrect analysis (excluding confounding variables) cannot lead to a correct conclusion.

6) Figure 2C shows a very interesting and productive result, that the difference in translation efficiency between the wild-type "C" ribosomes and the A-ribosomes correlates to some degree with the hybridization free energy between the mRNA and (a portion of) the anti-SD sequence. This is a productive approach towards eliminating key confounding variables because, in principle, the strengths of the four other interactions that control translation initiation rate should not change when the 16S rRNA aSD sequences are changed. However, it's not apparent in the manuscript text, but the authors are using the modified 16S rRNAs to “eliminate the SD-aSD interaction as a contribution to the mRNA's translation rate”. So when they subtract the contribution from the modified A-ribosome's translation rates from the C-ribosome's translation rate, they are observing more directly the contribution from the SD-aSD interaction. The manuscript text should more clearly explain this experimental design. This is a creative and valid way of using ribosome profiling measurements.

7) However, the hybridization free energy calculations could be improved. First, as mentioned previously, the wild-type aSD sequence in *E. coli* is ACCUCCUUA. Second, the hybridization free energy calculation was only performed on the region from 15 to 6 nucleotides upstream of the start codon, but the aSD sequence can hybridize at other locations. Third, the hybridization between the mRNA and aSD can accommodate 1 or 2-nucleotide bulges or internal loops.

8) The measurements in cold shock are greatly confounded by the higher expression levels of RNA chaperones that are unfolding mRNA structures “at specific mRNAs” where the RNA chaperones recognize binding motifs. The conclusion here should be that RNA chaperones bind specific mRNAs, unfold their inhibitory mRNA structures, and increase their translation rates during cold shock. This is all independent of the Shine-Dalgarno sequence. This process also does not depend on many other uninteresting factors.

9) The use of ORF-wide GINI values is odd because it's generally only the region surrounding the start codon that affects its translation initiation rate, and not the structure of the entire ORF (which this coefficient is quantifying). Also, using the SHAPE reactivity around a start codon as a proxy for RNA structure is a bit misleading as ribosomes actively unfold RNA structures during translation initiation. A highly structured mRNA with a consensus SD sequence will have a high SHAPE reactivity (i.e., low RNA structure) because the ribosomes can rapidly bind to the mRNA and unfold the mRNA structure. SHAPE reactivity is measuring the effect of rapid ribosome binding and not the cause of it. Rapid ribosome binding can also be facilitated by slow RNA refolding kinetics, called "Ribosome Drafting" in the literature.

10) The data in Figure 6 just says that A-ribosomes can initiate translation rate at other start codons because they now have more negative binding free energies to those start codons, compared to the annotated ones. The authors could perform hybridization calculations using the A-ribosome's aSD sequence to investigate whether these "new start codons" have a nearby "SD" sequence that is complementary to the A-ribosome's aSD. That would be interesting.

[Editors’ note: further revisions were suggested prior to acceptance, as described below.]

Thank you for resubmitting your work entitled "Translational initiation in *E. coli* occurs at the correct sites genome-wide in the absence of mRNA-rRNA base-pairing" for further consideration by *eLife*. Your revised article has been evaluated by James Manley as the Senior Editor, and three reviewers, including Joe Wade as the Reviewing Editor and Reviewer #1.

The reviewers and editors agree that the revised manuscript is greatly improved, and we are pleased to provisionally accept the manuscript. We ask that you make a few small changes in response to the reviewers' comments. First, reviewer 2 has two minor concerns that are easily addressed. Second, based on reviewer 3's comments, the conclusions regarding the importance of A-rich sequences should be softened a little. The reviewers' comments are listed below:

Reviewer #1:

The authors have done an excellent job improving the manuscript. Removing the data on cold-shock has improved the focus and readability. Moreover, the new analyses in Figures 3 and 4 make a more compelling case that Shine-Dalgarno sequences are neither necessary nor sufficient for start site selection.

Reviewer #2:

The streamlined paper of Saito et al. reads much better than the original version and delivers a clear and impactful message.

I believe it can be published after authors address two remaining issues:

The authors refer to "the number of elongating ribosomes per mRNA as a proxy of initiation rates". This is incorrect: there would be twice as many ribosomes on an mRNA that is twice as long as another one, even if those two would have the same initiation rate. The correct metrics is not the number of ribosomes per mRNA but the ribosome density (their number normalized by mRNA length). This does not affect conclusions of the paper because authors normalize RiboSeq reads by RNASeq reads. Yet, I would try to avoid this confusion.

The authors write: "Interestingly, in comparing internal AUG codons that support initiation in our ribosome profiling data to those that do not, we found that A's are enriched both upstream and downstream of initiation sites (Figure 5A).…. This results from endogenous initiation sites… ". However, Figure 5A does *not* deal with the *internal* initiation sites, but with the annotated sites lacking SD.

Reviewer #3:

With the revisions, the authors have greatly improved the manuscript's introductory description of translation and the overall analysis of their dataset, leading to a more laser-focused and well-supported set of conclusions. These results provide an excellent and clarifying view of the sequence determinants and interactions that control translation initiation rate within natural (highly evolved) mRNAs by cleanly separating the role of the SD:aSD interaction from other factors, including the presence/absence of inhibitory mRNA structures.

The Introduction provides a more comprehensive description of the several sequence determinants and factors that control translation initiation rate, which is essential towards understanding the authors' excellent dataset. The analysis clearly explains how their dataset provides comparative measurements with vs. without the SD:aSD interaction and how those measurements quantify its effect on translation rate. Start codon selection is a property of all the factors that control translation initiation rate, and is also likely a property of ribosome-ribosome dynamics along the mRNA.

---

## [Author Response]

[Editors’ note: the authors resubmitted a revised version of the paper for consideration. What follows is the authors’ response to the first round of review.]

Reviewer #1:This paper describes an innovative approach to probe the importance of Shine-Dalgarno (S-D) sequences in translation initiation in *Escherichia coli*. […] I also have concerns about the interpretation of Figures 6 and 7 that impact the overall conclusions.- The presentation of the cold shock data is confusing. The overall conclusion is that S-D-dependence is lost at almost all genes during cold shock. However, a small subset of genes appears to depend strongly on the S-D. The distinction between the effect on the majority of genes and the effect on a small subset should be explained more clearly. The simplest interpretation of these data is that most start codons are highly structured during cold shock, but those that are do not rely on their S-Ds. This model is largely consistent with previous work.

We have removed the figures dealing with cold shock and other stresses. We agree that they are largely consistent with previous work. The confusion raised by the cold shock story appears to have taken away from the main story.

- I disagree with the interpretation of Figure 6. The data show that for the altered ribosomes, annotated start codons are used far more efficiently than the collection of all other ATG sequences within ORFs. However, there are many more ATGs within ORFs than annotated start codons, and even if translation relies heavily on S-D sequences, you would expect that most ATGs within ORFs would not be selected by alternative ribosomes because only a small subset will have appropriate S-D sequences, and many may be weakly expressed. My interpretation of these data is that alternative ribosomes do use annotated start codons, but there is no way to tell how selectively they do this. A more appropriate comparison would be of (i) annotated start codons to (ii) ATGs within ORFs where the ATG is associated with a sequence that is predicted to function as a good S-D for the alternative ribosome.

We have added analyses to the new Figure 4 and Figure 4—figure supplement 1 showing initiation at internal AUG codons predicted to have high affinity for the mutant ASD sequences. These data show that initiation occurs with all four ribosome types regardless of the SD strength or specificity, but that initiation is most efficient when the SD and ASD are complementary.

- Another concern I have with Figure 6 is that presumably some, and perhaps many of the annotated start codons will have good SD matches for the alternative ribosomes. Figure 2C suggests that the number with good matches will be fairly high. Is the ribosome density at annotated start codons simply due to the subset of start codons that have reasonable SD matches to the altered ASD? Another way to think about this is to ask whether the start codons contributing to the signal in Figure 6C are the same start codons that contribute to the signal in Supplementary Figure 5A-B.

We added analyses to the new Figure 3 showing that initiation occurs with all three mutant ribosomes at annotated start sites that have no affinity for the mutant ASD sequences.

- Figure 7E shows the importance of an A-rich sequence in the context of start codons lacking a good S-D. Similar to Figure 6, these data highlight the contribution of non-SD sequences to translation initiation, but they do not provide any information about the relative importance of the different sequence elements.

We have removed claims about the relative importance of different sequence elements.

Reviewer #2:Critique:This is an interesting, intriguing and important study. The results are nice and clean and the implications are important for unraveling the fundamental mechanism of translation initiation in bacteria. Although the paper is generally well written, it was hard at times to follow the authors logic and I strongly encourage the authors to try to clarify the message, which often was hard to extract.1) Here are several examples:- Abstract: The statement "We reveal a genome-wide correlation between the SD strength and translational efficiency" is followed by "this global correlation is lost and a subset of genes […] becomes [dependent] on SD motifs for translation". This is hard to digest.- Figure 4C legend ("the strength of the SD motifs determines whether wild-type or ASD mutants are recruited to messages") is supposed to contrast Figure 4F legend ("the unstructured SD motifs can recruit wild type ribosomes more effectively than they recruit ASD mutants"). However, they sound nearly identical and thus, do not accurately communicate the point the authors apparently are trying to make.- "genes with strong SD motif are translated better by ribosomes with canonical ASD": better in comparison with the ASD-mutant ribosomes or better in comparison with the genes with weak SD?

We removed the section on stress conditions that was hard to follow and taking away from the main point of the manuscript.

2) Aleksashin et al., 2019, have shown that altering ASD in 16S rRNA compromises rRNA maturation. Although the presence of unprocessed sequences at the 5' and 3' end of the ASD-mutant 16S rRNA would not likely change the general conclusions of the paper, hypothetically it could affect the functionality and elongation rate of the mutant ribosomes. I am wondering whether authors have checked how well their mutant 16S rRNAs are processed. Irrespectively, I believe a more detailed discussion of the general functionality of the ribosomes with altered ASD, especially in relation to the elongation rate, would be beneficial.

RNA-seq analyses of rRNA (prior to nuclease treatment) is now shown in Figure 1—figure supplement 1 and discussed early in the Results section (subsection “Selective profiling of ribosomes with mutant ASD sequences”).

3) Subsection “Gene-specific roles of SD motifs under stress”. The readers need a better explanation why the authors switched from ΔlogTE to ΔlogRPKM metrics when they move to the experiments in the stressed cells.

This section was removed.

4) The influence of the competition between wt and mutant 30S subunits for the translation start sites on the conclusions drawn from ribosome profiling should be discussed.

This possibility was added to the Discussion.

Reviewer #3:1) The author focuses primarily on the importance of the sequence colloquially known as the Shine-Dalgarno in controlling a mRNA's translation initiation rate. The authors write that "Initiation rates vary depending on how well an mRNA recruits 30S subunits to the start codon, and in bacteria, the working model is that this is accomplished primarily by Shine-Dalgarno (SD) motifs." This is incorrect. […] Their current conclusions are already subsumed within the state-of-the-art (i.e., nothing new).

We added a more detailed description of the factors that affect initiation rates to the Introduction, including the points listed above.

2) Any discussion of "which translation rate interaction is most important" or "which translation rate interaction is responsible for X whereas the interaction Y only fine-tunes Z" is not productive and can easily be contradicted by selecting a real counter example. Overall, it is the binding free energy of the 30S ribosome to the mRNA that determines its translation initiation rate. Each of these interactions contributes free energy to this process and the magnitude of the contributed free energies can be roughly equal across a selection of real mRNA examples. There are unstructured mRNAs where there is little penalty for unfolding inhibitory mRNA structures. There are highly structured mRNAs that have consensus SDs sequences. There are mRNAs that have consensus SD sequences far away from the start codon. All of these mRNAs could have the same translation rate. Which interaction is most important? That's not the right question to ask, because it's meaningless.

We have removed language that focuses on the relative contribution of the individual factors that affect translational initiation. We agree that our analyses do not allow us to determine their relative contributions.

3) The manuscript's main topic is the Shine-Dalgarno sequence, but the authors should be made aware that at least the last 9 nucleotides of the 16S rRNA can contact the mRNA and hybridize to it. In *E. coli*, the anti-Shine Dalgarno sequence is 5'-ACCUCCUUA-3' and the "consensus" Shine-Dalgarno sequence is therefore 5'-TAAGGAGGT-3'. The manuscript text and the authors' calculations should reflect this.

We revised the Introduction to explicitly state that up to 9 bp can form. Drawing on previous work, we refer to the “consensus” as GGAGG because it is the G’s that are overrepresented upstream of start codons (see the data for *E. coli* in Figure 5—figure supplement 1).

4) The authors are mis-using the ribosome profiling measurements in their analysis. Ribosome profiling measurements do not directly measure translation rates. They measure mRNA-bound ribosome densities. A mRNA's ribosome density will depend on both its translation initiation rate AND its translation elongation rate. Specifically, in steady-state conditions, the ribosome density will be the ratio between these two quantities (initiation rate over elongation rate). In the initial applications of ribosome profiling, researchers assumed that all mRNAs have the same translation elongation rate in order to conclude that ribosome density measurements were proportional to translation initiation rates. This is not true. Coding sequences in mRNAs have very different translation elongation rates, due to differences in synonymous codon usage. Unless each mRNAs' translation elongation rates are predicted or directly measured, ribosome density measurements cannot be used to infer their translation initiation rates. Therefore, when the authors write "In pioneering ribosome profiling studies in bacteria, the paradoxical observation was made that there is little or no correlation between the translational efficiency of a gene and the strength of its SD motif (calculated using thermodynamic algorithms for RNA pairing), as had been anticipated based on the SD model." there is no actual paradox. The ribosome profiling measurements were not used correctly to test how mRNA sequences control translation rate.

We revised the Introduction to explicitly state that up to 9 bp can form. Drawing on previous work, we refer to the “consensus” as GGAGG because it is the G’s that are overrepresented upstream of start codons (see the data for *E. coli* in Figure 5—figure supplement 1).

5) Getting to the authors' main conclusions, they write that "These data indicate that the ASD mutant ribosomes translate genes with weak SD motifs better than genes with strong SD motifs, exactly the opposite of what wild-type ribosomes are expected to do." This statement is confusing given the real conclusion of the authors, that all other factors being equal a "strong" SD motif does result in higher translation than a "weak" SD motif. It's only because of other confounding factors that the initial analysis did not yield a positive correlation. An incorrect analysis (excluding confounding variables) cannot lead to a correct conclusion.

The sentence, “These data indicate that the ASD mutant ribosomes translate genes with weak SD motifs better than genes with strong SD motifs” describes the observations in Figure 2B, the result of all the other factors except for SD-ASD pairing. We removed the phrase “exactly the opposite of what wild-type ribosomes are expected to do” that seems to have caused the confusion.

6) Figure 2C shows a very interesting and productive result, that the difference in translation efficiency between the wild-type "C" ribosomes and the A-ribosomes correlates to some degree with the hybridization free energy between the mRNA and (a portion of) the anti-SD sequence. This is a productive approach towards eliminating key confounding variables because, in principle, the strengths of the four other interactions that control translation initiation rate should not change when the 16S rRNA aSD sequences are changed. However, it's not apparent in the manuscript text, but the authors are using the modified 16S rRNAs to “eliminate the SD-aSD interaction as a contribution to the mRNA's translation rate”. So when they subtract the contribution from the modified A-ribosome's translation rates from the C-ribosome's translation rate, they are observing more directly the contribution from the SD-aSD interaction. The manuscript text should more clearly explain this experimental design. This is a creative and valid way of using ribosome profiling measurements.

We revised the language at the end of the Introduction and the beginning of the Results section to better explain our experimental design. The fact that we can isolate the effects of SD-ASD interactions from the other factors that set initiation rates explains why we use statistics for single variables and focus primarily on the SD mechanism of initiation.

7) However, the hybridization free energy calculations could be improved. First, as mentioned previously, the wild-type aSD sequence in *E. coli* is ACCUCCUUA. Second, the hybridization free energy calculation was only performed on the region from 15 to 6 nucleotides upstream of the start codon, but the aSD sequence can hybridize at other locations. Third, the hybridization between the mRNA and aSD can accommodate 1 or 2-nucleotide bulges or internal loops.

The fact that we see a strong correlation between our calculated SD affinities and differences in ribosome occupancy (WT – mutant) argues that the calculations are basically reliable. We see the highest correlation when affinities are calculated using the 10 nt between -15 and -6 from the AUG and we show the data for various windows with different SD distances in Figure 2—figure supplement 2. The calculations are quite robust to changes in parameters: we see little or no differences in SDRO correlations if we use 9 nt of ASD sequence to calculate affinities instead of 7 nt, or if we allow the ASD to pair anywhere between -20 to 0 upstream of AUG, or if we use the RBS calculator to generate the ΔG values.

8) The measurements in cold shock are greatly confounded by the higher expression levels of RNA chaperones that are unfolding mRNA structures “at specific mRNAs” where the RNA chaperones recognize binding motifs. The conclusion here should be that RNA chaperones bind specific mRNAs, unfold their inhibitory mRNA structures, and increase their translation rates during cold shock. This is all independent of the Shine-Dalgarno sequence. This process also does not depend on many other uninteresting factors.

We removed the section on the role of SD motifs under stress because it generated confusion and ornithological references without strengthening the main point of our manuscript.

9) The use of ORF-wide GINI values is odd because it's generally only the region surrounding the start codon that affects its translation initiation rate, and not the structure of the entire ORF (which this coefficient is quantifying). Also, using the SHAPE reactivity around a start codon as a proxy for RNA structure is a bit misleading as ribosomes actively unfold RNA structures during translation initiation. A highly structured mRNA with a consensus SD sequence will have a high SHAPE reactivity (i.e., low RNA structure) because the ribosomes can rapidly bind to the mRNA and unfold the mRNA structure. SHAPE reactivity is measuring the effect of rapid ribosome binding and not the cause of it. Rapid ribosome binding can also be facilitated by slow RNA refolding kinetics, called "Ribosome Drafting" in the literature.

Carol Gross and colleagues showed that ORF-wide GINI values are highly correlated with translational efficiency genome-wide. This is true whether the DMS probing is done in vivo (where ribosomes could affect structure by unwinding the RNA) or to a lesser extent with purified RNA in vitro. Kevin Weeks also showed that RNA structures are correlated in vivo and in vitro using the SHAPE reagent. These data argue that at least to some extent mRNA structure is driving translation rates. This was clarified in the Results section in the discussion of Figure 5E.

10) The data in Figure 6 just says that A-ribosomes can initiate translation rate at other start codons because they now have more negative binding free energies to those start codons, compared to the annotated ones. The authors could perform hybridization calculations using the A-ribosome's aSD sequence to investigate whether these "new start codons" have a nearby "SD" sequence that is complementary to the A-ribosome's aSD. That would be interesting.

The new Figures 3 and 4 now include analyses of sets of initiation sites with various affinities for the WT or mutant ASD sequences showing more clearly that translation occurs at start codons even without strong SD-ASD pairing.

[Editors’ note: what follows is the authors’ response to the second round of review.]

Reviewer #2:The streamlined paper of Saito et al. reads much better than the original version and delivers a clear and impactful message.I believe it can be published after authors address two remaining issues:The authors refer to "the number of elongating ribosomes per mRNA as a proxy of initiation rates". This is incorrect: there would be twice as many ribosomes on an mRNA that is twice as long as another one, even if those two would have the same initiation rate. The correct metrics is not the number of ribosomes per mRNA but the ribosome density (their number normalized by mRNA length). This does not affect conclusions of the paper because authors normalize RiboSeq reads by RNASeq reads. Yet, I would try to avoid this confusion.

The language was changed to “ribosome density” instead of “the number of ribosomes.”

The authors write: "Interestingly, in comparing internal AUG codons that support initiation in our ribosome profiling data to those that do not, we found that A's are enriched both upstream and downstream of initiation sites (Figure 5A).…. This results from endogenous initiation sites… ". However, Figure 5A does not deal with the internal initiation sites, but with the annotated sites lacking SD.

Thanks for catching this mistake; the Discussion was updated to reflect this.